# AEDESIGN: A GRAPH PROTEIN DESIGN METHOD AND BENCHMARK ON ALPHAFOLD DB

## ABSTRACT

While AlphaFold has remarkably advanced protein folding, the inverse problem, protein design, by which protein sequences are predicted from the corresponding 3D structures, still faces significant challenges. First of all, there lacks a large-scale benchmark covering the vast protein space for evaluating methods and models; secondly, existing methods are still low in prediction accuracy and time-inefficient inference. This paper establishes a new benchmark based on AlphaFold DB, one of the world's largest protein structure databases. Moreover, we propose a new baseline method called AEDesign, which achieves 5% higher recovery than previous methods and about 70 times inference speed-up in designing long protein sequences. We also reveal AEDesign's potential for practical protein design tasks, where the designed proteins achieve good structural compatibility with native structures. The open-source code will be released.

## 1 INTRODUCTION

As "life machines", proteins play vital roles in almost all cellular processes, such as transcription, translation, signaling, and cell cycle control. Understanding the relationship between protein structures and their sequences brings significant scientific impacts and social benefits in many fields, such as bioenergy, medicine, and agriculture (Huo et al., 2011; Williams et al., 2019). While AlphaFold2 has tentatively solved protein folding (Jumper et al., 2021; Wu et al., 2022; Lin et al., 2022; Mirdita et al., 2022; Li et al., 2022c) from 1D sequences to 3D structures, its reverse problem, i.e., protein design raised by (Pabo, 1983) that aims to predict amino acid sequences from known 3D structures, has fewer breakthroughs in the ML community. The main reasons hindering the research progress include: (1) The lack of large-scale standardized benchmarks; (2) The difficulty in improving protein design accuracy; (3) Many methods are neither efficient nor open-source. Therefore, we aim to benchmark protein design and develop an effective and efficient open-source method.

Previous benchmarks may suffer from biased testing and unfair comparisons. Since SPIN (Li et al., 2014) introduced the TS500 (and TS50) consisting of 500 (and 50) native structures, it has served as a standard test set for evaluating different methods (O'Connell et al., 2018; Wang et al., 2018; Chen et al., 2019; Jing et al., 2020; Zhang et al., 2020a; Qi & Zhang, 2020; Strokach et al., 2020). However, such a few proteins do not cover the vast protein space and are more likely to lead to biased tests. Besides, there are no canonical training and validation sets, which means that different methods may use various training sets. If the training data is inconsistent, how can we determine that the performance gain comes from different methods rather than biases of the data distribution? Especially when the test set is small, adding training samples that match the test set distribution could cause dramatic performance fluctuations. Considering these issues, we suggest establishing a large-scale standardized benchmark for fair and comprehensive comparisons.

Extracting expressive residue representations is a key challenge for accurate protein design, where both sequential and structural properties must be considered. For general 3D points, structural features should be rotationally and translationally invariant in the classification task. Regarding proteins, we should consider amino acids' stable structure, number, and order. Previous studies (O'Connell et al., 2018; Wang et al., 2018; Ingraham et al., 2019; Jing et al., 2020) may have overlooked some important protein features and data dependencies, i.e., bond angles; thus, few of them exceeds 50% recovery except DenseCPD (Qi & Zhang, 2020). How can protein features and neural models be designed to learn expressive residue representations?

Improving the model efficiency is necessary for rapid iteration of research and applications. Current advanced methods have severe speed defects due to the sequential prediction paradigm. For example, GraphTrans (Ingraham et al., 2019) and GVP (Jing et al., 2020) predict residues one by one during inference rather than in parallel, which means that it calls the model multiple times to get the entire

protein sequence. Moreover, DenseCPD (Qi & Zhang, 2020) takes 7 minutes to predict a 120-length protein on their server [1]. How can we improve the model efficiency while ensuring accuracy?

To address these problems, we establish a new protein design benchmark and develop a graph model called AEDesign (Accurate and Efficient Protein Design) to achieve SOTA accuracy and efficiency. Firstly, we compare various graph models on consistent training, validation, and testing sets, where all these datasets come from the AlphaFold Protein Structure Database (Varadi et al., 2021). In contrast to previous studies (Ingraham et al., 2019; Jing et al., 2020) that use limited-length proteins, we extend the experimental setups to the case of arbitrary protein length. Secondly, we improve the model accuracy by introducing protein angles as new features and introducing a simplified graph transformer encoder (SGT). Thirdly, we improve model efficiency by proposing a confidence-aware protein decoder (CPD) to replace the auto-regressive decoder. Experiments show that AEDesign significantly outperforms previous methods in accuracy (+5%) and efficiency (70+ times faster than before). We also reveal AEDesign's potential for practical protein design tasks, where the designed proteins achieve good structural compatibility with native structures.

## 2 RELATED WORK

We focus on structure-based protein design (Gao et al., 2020; Pearce & Zhang, 2021; Wu et al., 2021; Ovchinnikov & Huang, 2021; Ding et al., 2022; Strokach & Kim, 2022; Li & Koehl, 2014; Greener et al., 2018; Anand et al., 2022; Karimi et al., 2020; Cao et al., 2021; Liu et al., 2022; McPartlon et al., 2022; Huang et al., 2022; Dumortier et al., 2022; Li et al., 2022a; Maguire et al., 2021; Anishchenko et al., 2021; Li et al., 2022b), and the approaches can be categorized into MLP-based, CNN-based, and GNN-based ones. Some terms need to be explained in advance: we refer to amino acids as residues, and accuracy indicates the degree of prediction of the residues, i.e., recovery.

**Problem definition** The structure-based protein design aims to find the amino acids sequence $\mathcal{S} = \{s_i : 1 \leq i \leq n\}$ that folds into a known 3D structure $\mathcal{X} = \{\boldsymbol{x}_i \in \mathbb{R}^3 : 1 \leq i \leq n\}$, where $n$ is the number of residues and the natural proteins are composed by 20 types of amino acids, i.e., $1 \leq s_i \leq 20$, and $s_i \in \mathbb{N}^+$. Formally, that is to learn a function $\mathcal{F}_\theta : \mathcal{X} \mapsto \mathcal{S}$. Because homologous proteins always share similar structures (Pearson & Sierk, 2005), the problem itself is underdetermined, i.e., the valid amino acid sequence may not be unique. In addition, the need to consider both 1D sequential and 3D structural information further increases the difficulty of algorithm design.

**Table 1:** Statistics of structure-based protein design methods. $N_{train}$ is the number of training samples. TS500 and TS500 are the test sets containing 500 and 50 proteins, respectively. All results are copied from their manuscripts or related papers.

| | Method | $N_{train}$ | TS500 | TS50 | Code |
|---|---|---|---|---|---|
| MLP | SPIN | 1,532 | 30.30% | 30.30% | no |
| | SPIN2 | 1,532 | 36.60% | 33.60% | no |
| | Wang's model | 10,173 | 36.14% | 33.00% | no |
| CNN | SPROF | 7,134 | 40.25% | 39.16% | PyTorch |
| | ProDCoNN | 17,044 | 42.20% | 40.69% | no |
| | DenseCPD | ≤10,727 | 55.53% | 50.71% | no |
| GNN | GraphTrans | 18,024 | – | – | PyTorch |
| | GVP | 18,024 | – | 44.90% | PyTorch |
| | GCA | 18,024 | – | 43.00% | PyTorch |
| | AEDesign | 18,024 | 49.23% | 48.36% | PyTorch |
| | ESM-IF | >1M | – | – | PyTorch |
| | ProteinMPNN | 18,024 | – | – | PyTorch |

**MLP-based models** These methods use **m**ulti-**l**ayer **p**erceptron (MLP) to predict the type of each residue. The MLP outputs the probability of 20 amino acids for each residue, and the input feature construction is the main difference between various methods. SPIN (Li et al., 2014) integrates torsion angles ($\phi$ and $\psi$), fragment-derived sequence profiles, and structure-derived energy profiles to predict protein sequences. SPIN2 (O'Connell et al., 2018) adds backbone angles ($\theta$ and $\tau$), local contact number, and neighborhood distance to improve the accuracy from 30% to 34%. (Wang et al., 2018) uses backbone dihedrals ($\phi$, $\psi$ and $\omega$), the solvent accessible surface area of backbone atoms ($C_\alpha, N, C$, and $O$), secondary structure types (helix, sheet, loop), $C_\alpha - C_\alpha$ distance and unit direction vectors of $C_\alpha - C_\alpha$, $C_\alpha - N$ and $C_\alpha - C$, which achieves 33.0% accuracy on 50 test proteins. The MLP methods have a high inference speed, but their accuracy is relatively low due to the partial consideration of structural information. These methods require complex feature engineering using multiple databases and computational tools, limiting their widespread usage.

**CNN-based models** CNN methods extract protein features directly from the 3D structure (Torng & Altman, 2017; Boomsma & Frellsen, 2017; Weiler et al., 2018; Zhang et al., 2020a; Huang et al., 2017; Chen et al., 2019), which can be further classified as 2D CNN-based and 3D CNN-based.

---

[1] http://protein.org.cn/densecpd.html

The 2D CNN-based SPROF (Chen et al., 2019) extracts structural features from the distance matrix and improves the accuracy to 39.8%. In contrast, 3D CNN-based methods extract residue features from the atom distribution in a three-dimensional grid box. For each residue, the atomic density distribution is computed after being translated and rotated to a standard position so that the model can learn translation and rotation invariant features. ProDCoNN (Zhang et al., 2020a) designs a nine-layer 3D CNN to predict the corresponding residues at each position, which uses multi-scale convolution kernels and achieves 40.69% recovery on TS50. DenseCPD (Qi & Zhang, 2020) further uses the DensetNet architecture (Huang et al., 2017) to boost the accuracy to 50.71%. Although 3DCNN-based models improve accuracy, their inference is slow, probably because they require separate pre-processing and prediction for each residue.

**Graph-based models** Graph methods represent the 3D structure as a $k$-NN graph, then use graph neural networks (Defferrard et al., 2016; Kipf & Welling, 2016; Veličković et al., 2017; Zhou et al., 2020; Zhang et al., 2020b; Gao et al., 2022; Li et al., 2021a) to learn residue representations considering structural constraints. The protein graph encodes the residue information in node vectors and constructs edges and edge features between neighboring residues. GraphTrans (Ingraham et al., 2019) combines graph encoder and autoregressive decoder to generate protein sequences. GVP (Jing et al., 2020) increases the accuracy to 44.9% by proposing the geometric vector perceptron, which learns both scalar and vector features in an equivariant and invariant manner concerning rotations and reflections. GCA (Tan et al., 2022) further improves recovery to 47.02% by introducing global graph attention. Another related work is ProteinSolver (Strokach et al., 2020), but it was mainly developed for scenarios where partial sequences are known and do not report results on standard datasets. In parallel with our work, ProteinMPNN (Dauparas et al., 2022) and ESM-IF (Hsu et al., 2022) achieve dramatic improvements, while they do not provide the training code and will be benchmarked in the future.

## 3 METHODS

### 3.1 OVERVIEW

We present the overall framework of AEDesign in Fig. 1. We suggest using AEDesign as a future baseline model because it is more accurate, straightforward, and efficient than previous methods. The methodological innovations include:

- **Expressive features**: We add new proteins angles $(\alpha, \beta, \gamma)$ to steadily improve the model accuracy.
- **Simplified graph encoder**: We use the **s**implified **g**raph **t**ransformer (SGT) to extract more expressive representations.
- **Fast sequence decoder**: We propose **c**onstraint-aware **p**rotein **d**ecoder (CPD) to speed up inference by replacing the autoregressive generator.

### 3.2 GRAPH FEATURE AND ENCODER

The protein structure can be viewed as a particular 3D point cloud in which the order of residues is known. For ordinary 3D points, there are two ways to get rotation and translation invariant features: Using particular network architectures (Fuchs et al., 2020; Satorras et al., 2021; Jing et al., 2020; Shuaibi et al., 2021) that take 3D points as input or using handicraft invariant features. For proteins, general 3D point cloud approaches cannot consider their particularity, including the regular structure and order of amino acids. Therefore, we prefer learning from the hand-designed, invariant, and protein-specific features. How to create invariant features and learn expressive representations from them is subject to further research.

**Graph** We represent the protein as a $k$-NN graph derived from residues to consider the 3D dependencies, where the default $k$ is 30. The protein graph $\mathcal{G}(A, X, E)$ consists of the adjacency matrix $A \in \{0, 1\}^{n,n}$, node features $X \in \mathbb{R}^{n,12}$, and edge features $E \in \mathbb{R}^{m,23}$. Note that $n$ and $m$ are the numbers of nodes and edges, and we create these features by residues' stable structure, order, and coordinates.

**Node features** As shown in Fig. 2, we consider two kinds of angles, i.e., the angles $\alpha_i, \beta_i, \gamma_i$ formed by adjacent edges and the dihedral angles $\phi_i, \psi_i, \omega_i$ formed by adjacent surfaces, where $\alpha_i, \beta_i, \gamma_i$ are new features we introduced. For better understanding dihedral angles, it is worth stating that $\phi_{i-1}$ is the angle between plane $C_{i-2} - N_{i-1} - C_{\alpha_{i-1}}$ and $N_{i-1} - C_{\alpha_{i-1}} - C_{i-1}$, whose intersection is $N_{i-1} - C_{\alpha_{i-1}}$. Finally, there are 12 node features derived from $\{\sin, \cos\} \times \{\alpha_i, \beta_i, \gamma_i, \phi_i, \psi_i, \omega_i\}$.

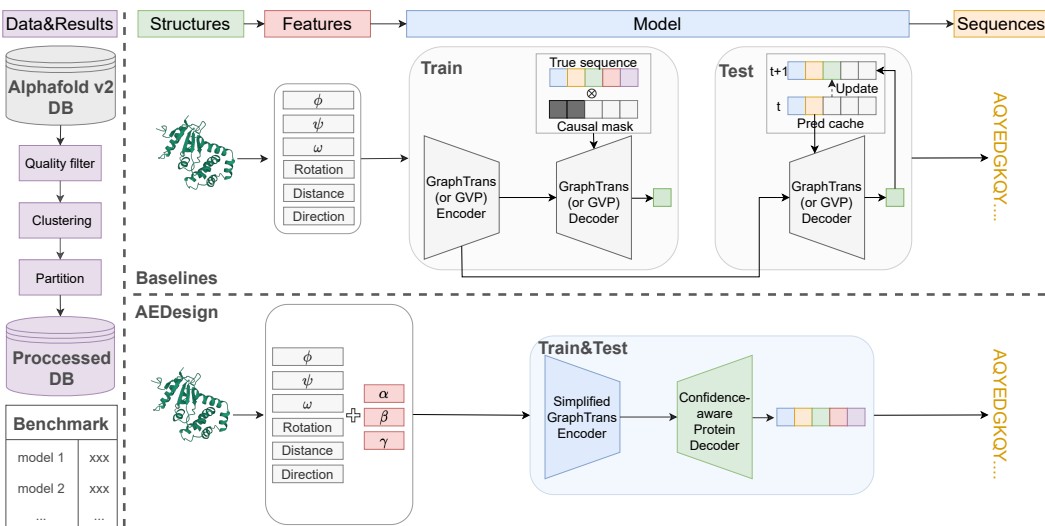

**Figure 1:** Overview of AEDesign. Compared with GraphTrans, StructGNN (Ingraham et al., 2019) and GVP (Jing et al., 2020), we add new protein features, simplify the graph transformer, and propose a confidence-aware protein decoder to improve accuracy and efficiency.

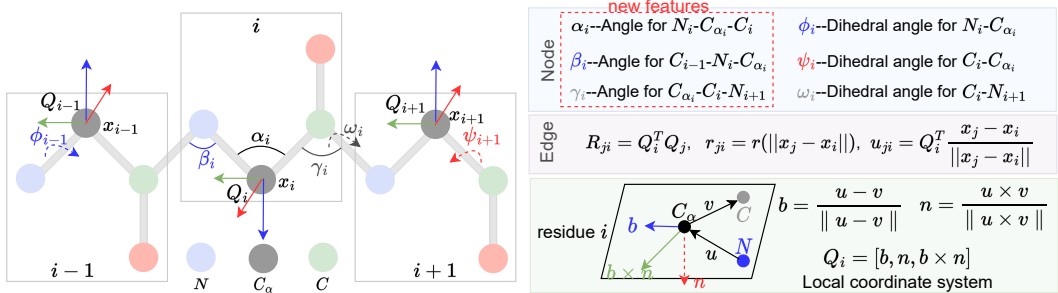

**Figure 2:** Angles of the protein backbone.

**Edge features** For edge $j \rightarrow i$, we use relative rotation $R_{ji}$, distance $r_{ji}$, and relative direction $u_{ji}$ as edge features, seeing Fig. 2. At the $i$-th residue's alpha carbon, we establish a local coordinate system $Q_i = (b_i, n_i, b_i \times n_i)$. The relative rotation between $Q_i$ and $Q_j$ is $R_{ji} = Q_i^T Q_j$ can be represented by an quaternion $q(R_{ji})$. We use radial basis $r(\cdot)$ to encode the distance between $C_{\alpha_i}$ and $C_{\alpha_j}$. The relative direction of $C_{\alpha_j}$ respective to $C_{\alpha_i}$ is calculated by $u_{ji} = Q_i^T \frac{x_j - x_i}{||x_j - x_i||}$. In summary, there are 23 edge features: 4 (quaternion) + 16 (radial basis) + 3 (relative direction).

**Simplified graph transformer** Denote $\boldsymbol{h}_i^l$ and $\boldsymbol{e}_{ji}^l$ as the output feature vectors of node $i$ and edge $j \rightarrow i$ in layer $l$. We use MLP to project input node and edge features into $d$-dimensional space, thus $\boldsymbol{h}_i^0 \in \mathbb{R}^d$ and $\boldsymbol{e}_{ji}^0 \in \mathbb{R}^d$. When considering the attention mechanisms centered in node $i$, the attention weight $a_{ji}$ at the $l + 1$ layer is calculated by:

$$\begin{cases} w_{ji} = MLP_1(\boldsymbol{h}_j^l || \boldsymbol{e}_{ji}^l || \boldsymbol{h}_i^l) \\ a_{ji} = \frac{\exp w_{ji}}{\sum_{k \in \mathcal{N}_i} \exp w_{ki}} \end{cases} \quad (1)$$

**Figure 3:** Simplified graph transformer.

where $\mathcal{N}_i$ is the neighborhood system of node $i$ and $||$ means the concatenation operation. Here, we simplify GraphTrans (Ingraham et al., 2019) by using a single MLP to learn multi-headed attention weights instead of using separate MLPs to learn $Q$ and $K$, seeing Fig. 3. The updated $\boldsymbol{h}_i^{l+1}$ is:

$$\begin{cases} \boldsymbol{v}_j = MLP_2(\boldsymbol{e}_{ji}^l || \boldsymbol{h}_j^l) \\ \boldsymbol{h}_i^{l+1} = \sum_{j \in \mathcal{N}_i} a_{ji} \boldsymbol{v}_j \end{cases} \tag{2}$$

By stacking multiple **s**implified **g**raph **t**ransformer (SGT) layers, we can obtain expressive protein representations considering 3D structural constraints by message passing.

### 3.3 SEQUENCE DECODER

To generate more accurate protein sequences, previous researches (Ingraham et al., 2019; Jing et al., 2020) prefer the autoregressive mechanism. However, this technique also significantly slows down the inference process because the residuals must be predicted one by one. *Can we parallelize the predictor while maintaining the accuracy?*

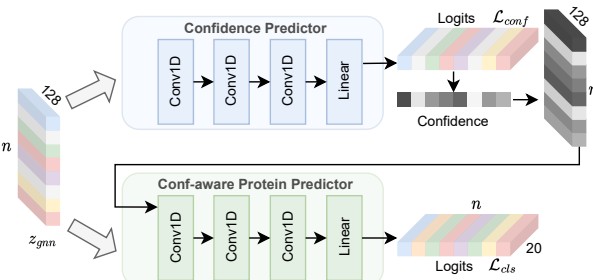

**Figure 4:** CPD: The confidence-aware protein decoder. We use two 1D CNN networks to learn confidence scores and make final predictions based on the input graph node features.

**Context-aware**    Previously, we have considered the 3D constraints through graph networks but ignored 1D inductive bias. As shown in Fig. 4, the input features are $\mathcal{Z}_{gnn} = \{z_1, z_2, \cdots, z_N\}$, where $z_i$ is the feature vector of node $i$ extracted by the encoder. In the generation phase, we use 1D CNNs to capture the local sequential dependencies based on the 3D context-aware graph node features, where the convolution kernel can be viewed as the sliding window.

**Confidence-aware**    Given the 3D structure $\mathcal{X} = \{x_i : 1 \leq i \leq N\}$ and protein sequence $\mathcal{S} = \{s_i : 1 \leq i \leq N\}$, the vanilla autoregressive prediction indicates $p(\mathcal{S}|\mathcal{X}) = \prod_i p(s_i|\mathcal{X}, s_{<i})$, where residues must be predicted one-by-one. We replace autoregressive connections with parallelly estimated confidence score $\boldsymbol{c}$, written as

$$\begin{cases} \boldsymbol{a} = \text{Conf}(\mathcal{X}) \\ \boldsymbol{c} = f(\boldsymbol{a}) \\ p(\mathcal{S}|\mathcal{X}) = \prod_i p(s_i|\mathcal{X}, x_i, c_i) \end{cases} \tag{3}$$

where $\text{Conf}(\cdot)$ is the model containing graph encoder and CNN decoder (called "Confidence predictor" in Figure.4) and outputs logit scores $\boldsymbol{a} \in \mathbb{R}^{n,20}$, $f(\cdot)$ is the function computing confidence score $\boldsymbol{c} \in \mathbb{N}^{n,1}$. The confidence score contains knowledge captured by the previous prediction and serves as a hint message to help the network correct previous predictions. Let $M = \text{ColumnMax}(\boldsymbol{a}) \in \mathbb{R}^{n,1}$ and $m = \text{ColumnSubMax}(\boldsymbol{a}) \in \mathbb{R}^{n,1}$ as the first and secondary largest logits score of $\boldsymbol{a}$, the confidence score is defined as $\boldsymbol{c} = \lfloor \frac{M}{m} \rfloor$. Let $\boldsymbol{a} \in \mathbb{R}^{1,20}$, and noting $i$ and $j$ as the indexes of the largest and sub-largest values of $\boldsymbol{a}$, then $M = a_i, m = a_j$, and $f(\boldsymbol{a}) = \lfloor \frac{a_i}{a_j} \rfloor$, where $\lfloor \cdot \rfloor$ indicates the floor function. By extending $\boldsymbol{a} \in \mathbb{R}^{1,20}$ as $\boldsymbol{a} \in \mathbb{R}^{n,20}$, Eq.(3) shows the vectorized version of $f(\cdot)$. We encode the confidence score as learnable embeddings $\mathcal{C} \in \mathbb{R}^{n,128}$, concatenate them with graph features, and feed these features into another CNN decoder to get the revised predictions. Note that all CNN decoders for estimating confidence and final predictions use the same CE loss:

$$\mathcal{L} = -\sum_i \sum_{1 \leq j \leq 20} \mathbb{1}_{\{j\}}(y_i) \log(p_{i,j}). \tag{4}$$

where $p_{i,j}$ is the predicted probability that residue $i$'s type is $j$, $y_i$ is the true label and $\mathbb{1}_{\{j\}}(\cdot)$ is a indicator function.

## 4 EXPERIMENTS

We conduct systematic experiments to establish a large-scale benchmark and evaluate the proposed AEDesign method. Specifically, we aim to answer:

- **Q1:** What is the difference between the new bechmark and the old one?
- **Q2:** Can AEDesign achieve SOTA accuracy and efficiency on the new benchmark?
- **Q3:** What is important for achieving SOTA performance?

### 4.1 BENCHMARK COMPARATION (Q1)

**Metric**   Following (Li et al., 2014; O'Connell et al., 2018; Wang et al., 2018; Ingraham et al., 2019; Jing et al., 2020), we use sequence recovery to evaluate different protein design methods. Compared with other metrics, such as perplexity, recovery is more intuitive and clear, and its value is equal to the average accuracy of predicted amino acids in a single protein sequence. By default, we report the median recovery score across the entire test set.

**Previous benchmark**   In Table. 1, we show the old benchmark collected from previous studies, including MLP (Li et al., 2014; O'Connell et al., 2018; Wang et al., 2018), CNN (Chen et al., 2019; Zhang et al., 2020a; Qi & Zhang, 2020; Huang et al., 2017) and GNN (Ingraham et al., 2019; Jing et al., 2020; Strokach et al., 2020; Dauparas et al., 2022; Hsu et al., 2022) models. Most approaches report results on the common test set TS50 (or TS500), consisting of 50 (or 500) native structures (Li et al., 2014). We also provide the results of AEDesign under the same experimental protocols as GraphTrans and GVP. Although the TS50, TS500 test sets have contributed significantly to establishing benchmarks, they still do not cover a vast protein space, and do not reveal how the model performs on species-specific data. Besides, there are no canonical training and validation sets, which means that different methods may use various training sets.

**New dataset**   We use the AlphaFold Protein Structure Database (until 2021.2.1) [2] (Varadi et al., 2021) to benchmark graph-based protein design methods. As shown in Table. 6 (Appendix), there are over 360,000 predicted structures by AlphaFold2 (Jumper et al., 2021) across 21 model-organism proteomes. This dataset has several advantages:

- Species-specific: This dataset provides well-organized species-specific data for different species, which is helpful to develop specialized models for each species.
- More structures: This dataset provides more than 360,000 structures, while Protein Data Bank (PDB) (Burley et al., 2021) holds just over 180,000 structures for over 55,000 distinct proteins.
- High quality: The median predictive score of AlphaFold2 reaches 92.4%, comparable to experimental techniques (Callaway, 2020). Nicholas (Fowler & Williamson, 2022) found that AlphaFold tends to be more accurate than NMR ensembles. In 2020, the CASP14 benchmark recognized AlphaFold2 as a solution to the protein–folding problem (Pereira et al., 2021).
- Missing value: There are no missing values in protein structures provided by AlphaFold DB.
- Widespread usage: AlphaFold DB has been used in many frontier works (Varadi et al., 2022; Morreale et al., 2022; Luyten et al., 2022; Alderson et al., 2022; Zhang et al., 2022; Fowler & Williamson, 2022; Shaban et al., 2022; Brems et al., 2022; Hsu et al., 2022), and we believe that investigating AlphaFold DB could yield more discoveries for protein design.

**Dataset Preprocessing**   It should be noted that the AlphaFold2 DB data itself may have model bias. Similar to ESM-IF (Hsu et al., 2022), we address data quality and partitioning issues through data pre-processing for each species-specific subset. As suggested by (Baek & Kepp, 2022; Callaway, 2020), the MAE between the predicted and experimentally generated structures does depend on pLDDT. Thus, we filter low-quality structures whose confidence score (pLDDT) is less than 70. To prevent potential information leakage, for each species-specific subset, we cluster protein sequences if their sequence similarities (Qi & Zhang, 2020; Steinegger & Söding, 2017) are higher than 30% (Qi & Zhang, 2020) and split the dataset by these clusters. As a result, the proteins belonging to the same cluster must be in one of the training, validation, and test sets. By default, we keep the

---

[2] https://alphafold.ebi.ac.uk

ratio of the training set and test set at about 9:1 and choose 100 proteins belonging to a randomly selected cluster for validation. If the randomly selected cluster has less than 100 proteins, then all of its proteins are used as the validation set.

## 4.2 AEDESIGN BENCHMARK (Q2)

**Overall settings** We extend the experimental setups to arbitrary length and species-specific, while most previous studies (Ingraham et al., 2019; Jing et al., 2020; Tan et al., 2022) do not explore such a vast protein space. Arbitrary length means that the protein length may be arbitrary to generalize the model to broader situations; otherwise, the protein must be between 30 and 500 in length. Species-specific indicates that we develop a specific model for each organism's proteome to learn domain-specific knowledge. In summary, there are two settings, i.e., species-specific dataset with limited length (SL) and species-specific dataset with arbitrary length (SA). Denote the total amount of structures as $N_{all}$, and the $i$-th species has $N_i$ structures, we have $N_{all} = \sum_{i=1}^{i=21} N_i$. As shown in Table. 6, if the length is limited, $N_{all} = 254,636$; otherwise, $N_{all} = 365,198$. For each species-specific subset, we develop 21 models based on datasets with $N_1, N_2, \cdots, N_{21}$ structures. We report the median recovery scores across the test set. All baseline results were obtained by running their official code with the same dataset. In parallel with our work, ProteinMPNN (Dauparas et al., 2022) and ESM-IF (Hsu et al., 2022) are also proposed, but they are not provided with the full training code and will be benchmarked in the future. Due to space constraints, we introduce abbreviations such as GTrans for GraphTrans and SGNN for Struct GNN.

**Hyper-parameters** AEDesign's encoder contains ten layers of SGT, and the decoder contains three layers of CNN, where the hidden dimension is 128. We use Adam optimizer and OneCycleLR scheduler to train all models up to 100 epochs with early stop patience 20 and learning rate 0.001. In the SL setting, we set the batch size as 16 for GraphTrans, StructGNN, GCA, and AEDesign, and the max node number as 2000 for GVP, which indicates the maximum number of residuals per batch. In the SA setting, we change GVP's max node parameter to 3000 to make it applicable to all data. GraphTrans, StructGNN, and GCA take more GPU memories because they must pad data according to the longest chain in each batch, and we adjust the batch size to 8 to avoid memory overflow.

**Table 2:** SL benchmark. The length of the protein must be between 30 and 500. We highlight the **best** (or next best) results in bold (or underline).

| | GTrans | SGNN | GVP | GCA | Our(SL) | Gain |
|---|---|---|---|---|---|---|
| DANRE | 44.93 | 55.82 | 62.16 | 63.23 | **70.70** | 7.47 |
| CANAL | 47.81 | 50.62 | 52.55 | 58.43 | **61.36** | 2.93 |
| MOUSE | 49.53 | 54.04 | 59.23 | 62.00 | **68.34** | 6.34 |
| ECOLI | 46.95 | 52.62 | 55.08 | 59.79 | **60.71** | 0.92 |
| DROME | 42.39 | 52.94 | 58.92 | 60.52 | **66.67** | 6.15 |
| METJA | 42.68 | 52.09 | 51.59 | 58.54 | **62.24** | 3.7 |
| PLAF7 | 46.32 | 50.00 | 49.52 | 58.74 | **61.16** | 2.42 |
| MYCTU | 54.02 | 56.32 | 58.50 | 65.24 | **68.68** | 3.44 |
| CAEEL | 47.44 | 54.89 | 62.32 | 63.00 | **70.23** | 7.23 |
| DICDI | 42.15 | 53.14 | 58.41 | 61.54 | **67.21** | 5.67 |
| TRYCC | 51.13 | 56.18 | 62.35 | 61.33 | **71.08** | 8.73 |
| YEAST | 36.18 | 49.23 | 52.14 | 58.30 | **62.50** | 4.2 |
| SCHPO | 44.56 | 49.34 | 51.90 | 58.21 | **60.64** | 2.43 |
| RAT | 48.23 | 60.41 | 67.01 | 65.87 | **75.50** | 8.49 |
| HUMAN | 51.15 | 54.83 | 61.19 | 63.10 | **69.69** | 6.59 |
| ARATH | 45.78 | 55.56 | 63.08 | 63.83 | **71.15** | 7.32 |
| MAIZE | 47.06 | 55.42 | 64.86 | 65.00 | **74.12** | 9.12 |
| LEIIN | 49.48 | 51.33 | 55.33 | 60.22 | **64.71** | 4.49 |
| STAA8 | 45.57 | 48.31 | 47.52 | 57.14 | **60.00** | 2.86 |
| SOYBN | 47.73 | 56.52 | 64.84 | 64.10 | **73.96** | 9.12 |
| ORYSJ | 46.29 | 54.62 | 63.64 | 63.64 | **72.98** | 9.34 |
| Average | 46.54 | 53.53 | 58.20 | 61.51 | **67.32** | 5.81 |

**Table 3:** SA benchmark. There is no constraint on the protein length. We highlight the **best** (or next best) results in bold (or underline).

| | GTrans | SGNN | GVP | GCA | Our(SA) | Gain |
|---|---|---|---|---|---|---|
| DANRE | 48.82 | 57.24 | 66.18 | 65.14 | **69.65** | 3.47 |
| CANAL | 51.14 | 55.17 | 61.82 | 60.88 | **68.45** | 6.63 |
| MOUSE | 47.79 | 57.40 | 64.23 | 64.50 | **74.07** | 9.57 |
| ECOLI | 50.60 | 54.22 | 59.11 | 60.71 | **66.22** | 5.51 |
| DROME | 50.56 | 55.91 | 63.13 | 61.67 | **71.55** | 8.42 |
| METJA | 51.03 | 52.47 | 53.13 | 59.24 | **59.64** | 0.4 |
| PLAF7 | 48.11 | 53.92 | 57.26 | 59.75 | **66.01** | 6.26 |
| MYCTU | 48.25 | 58.04 | 61.93 | 64.93 | **70.57** | 5.64 |
| CAEEL | 46.82 | 54.05 | 64.48 | 64.03 | **73.17** | 8.69 |
| DICDI | 55.58 | 56.82 | 64.17 | 62.91 | **71.66** | 7.49 |
| TRYCC | 48.56 | 59.57 | 64.52 | 66.06 | **75.18** | 9.12 |
| YEAST | 50.00 | 54.00 | 60.41 | 58.45 | **66.71** | 6.3 |
| SCHPO | 47.62 | 53.91 | 60.26 | 60.21 | **66.64** | 6.38 |
| RAT | 47.97 | 57.83 | 65.99 | 63.89 | **73.54** | 7.55 |
| HUMAN | 55.28 | 56.29 | 64.86 | 63.83 | **67.43** | 2.57 |
| ARATH | 48.39 | 57.59 | 66.55 | 64.66 | **73.84** | 7.29 |
| MAIZE | 56.76 | 58.57 | 67.78 | 65.83 | **76.00** | 8.22 |
| LEIIN | 54.58 | 56.85 | **63.92** | 62.81 | 63.54 | -0.38 |
| STAA8 | 49.36 | 51.43 | 53.85 | 58.46 | **63.85** | 5.39 |
| SOYBN | 49.94 | 58.44 | 66.25 | 69.23 | **76.51** | 7.28 |
| ORYSJ | 48.04 | 58.41 | 66.78 | 64.00 | **75.15** | 8.37 |
| Average | 50.24 | 56.10 | 62.70 | 62.91 | **69.97** | 7.06 |

**SL&SA results** We present the SL and SA benchmark in Table. 2 and Table. 3, respectively. The order of model accuracy from high to low is AEDesign > GCA > GVP > SGNN > GTrans. Under SL setting, AEDesign achieves the best recovery on all species-specific subsets and exceeds previous methods by 5.81% on average. Under SA setting, AEDesign is still the most accurate model, and the increased training data further improves the model, i.e., the average accuracy of AEDesign (SA) is 69.97% while that of AEDesign (SL) is 62.91%. For species-specific subsets, the larger the data

volume, the better model performance; refer to Table.6. For example, the recovery on METIJ is 58.54% when the number of structures is 1605, which will increase to 73.96% on SOYBN when the data volume increases to 41,048.

## 4.3 EFFICIENCY COMPARATION (Q2)

A good algorithm should have excellent computational efficiency in addition to high accuracy. We compare the inference time cost of different approaches, especially when designing long proteins commonly found in AF2DB.

**Setting** We evaluate various models' inference time costs under different scenarios, considering 100 proteins with short (L<500), medium (500<L<1000), and long (1000<L) lengths . As to long sequence design, we further investigate the time costs of encoder, decoder, and encoder+decoder. All experiments are conducted on an NVIDIA-V100.

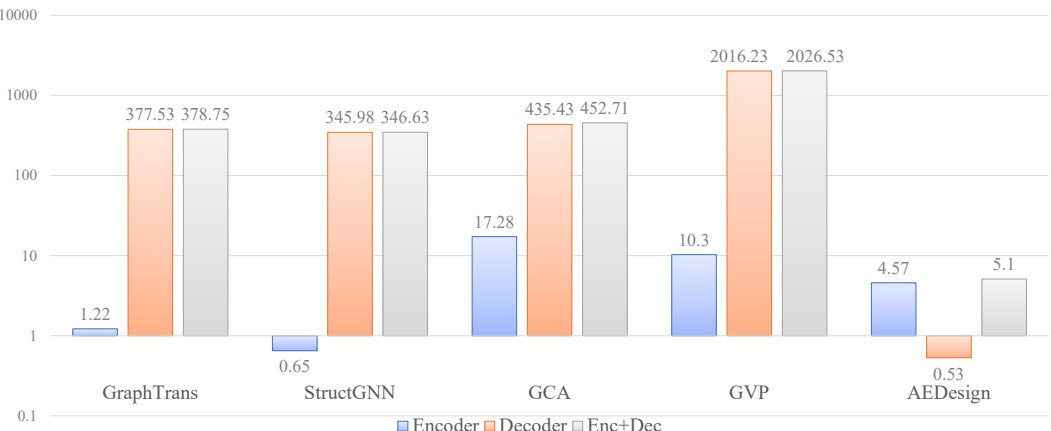

**Figure 5:** Inference time cost when designing long proteins. We report the total inference time of different methods on 100 long proteins, which are longer than 1000. The time costs of the encoder, decoder and encoder+decoder are reported.

**Table 4:** Inference time costs of different methods. #Number is the number of proteins for time cost evaluation. #Avg L is the average protein length.

|  | Short | Medium | Long |
|---|---|---|---|
| #Number | 100 | 100 | 100 |
| #Avg L | 286 | 724 | 1632 |
| GraphTrans | 47.58 | 262.87 | 378.76 |
| StructGNN | 43.63 | 213.87 | 346.63 |
| GCA | 44.99 | 230.64 | 452.71 |
| GVP | 208.85 | 969.51 | 2026.53 |
| AEDesign | **1.61** | **2.77** | **5.09** |

**Results** We show the inference time costs for designing proteins of different lengths in Table.4. When designing short proteins (L < 500), AEDesign is 25+ times faster than baselines; and the dominance extends to about 70 times for designing long proteins (L ≥ 500). In Fig. 5, we observe that the time costs of baselines mainly come from the autoregressive decoder, and the proposed CPD module significantly speeds up the decoding process.

## 4.4 ABLATION STUDY (Q3)

While AEDesign has shown remarkable performance, we are more interested in where the improvements come from. As mentioned before, we add new protein features, simplify the graph transformer, and propose a confidence-aware protein decoder. Whether these modifications could improve model performance?

**Setting** We conduct ablation studies under the SL setting. Specifically, we may replace the simplified attention module with the original GraphTrans (w/o SGT), replace the CPD module with the autoregressive decoder of GraphTrans (w/o CPD), or remove the newly introduced angle features (w/o new feat). All experimental settings keep the same as previous SL settings.

**Results and analysis** The ablation results are shown in Table. 5. We conclude that: (1) The SGT module and the new features can improve the recovery rate by 2.51% and 10.85%, respectively. Most of the performance improvement comes from the new angular features, are consistent with the recent ProteinMPNN, while they focus on distance features. (2) If we replace the CPD module with the autoregressive decoder, the recovery rate will improve by 0.55%. However, the recovery improve-

| | ADesign | w/o SGT | w/o CPD | w/o new feat |
|---|---|---|---|---|
| DANRE | 70.70 | 68.27 | 71.04 | 62.18 |
| CANAL | 61.36 | 59.00 | 62.53 | 53.45 |
| MOUSE | 68.34 | 65.88 | 68.70 | 60.00 |
| ECOLI | 60.71 | 58.27 | 61.05 | 52.97 |
| DROME | 66.67 | 64.22 | 67.11 | 54.45 |
| METJA | 62.24 | 59.23 | 63.10 | 50.61 |
| PLAF7 | 61.16 | 57.64 | 62.67 | 50.01 |
| MYCTU | 68.68 | 65.21 | 69.57 | 56.02 |
| CAEEL | 70.23 | 67.89 | 70.69 | 57.00 |
| DICDI | 67.21 | 63.79 | 67.48 | 54.94 |
| TRYCC | 71.08 | 68.93 | 71.70 | 59.58 |
| YEAST | 62.50 | 59.77 | 63.23 | 49.32 |
| SCHPO | 60.64 | 58.14 | 62.05 | 48.67 |
| RAT | 75.50 | 72.91 | 75.54 | 64.50 |
| HUMAN | 69.69 | 67.28 | 70.21 | 58.55 |
| ARATH | 71.15 | 69.30 | 71.82 | 61.62 |
| MAIZE | 74.12 | 72.17 | 74.42 | 64.26 |
| LEIIN | 64.71 | 61.98 | 65.29 | 52.52 |
| STAA8 | 60.00 | 57.87 | 60.33 | 47.31 |
| SOYBN | 73.96 | 72.06 | 73.58 | 63.89 |
| ORYSJ | 72.98 | 71.26 | 73.23 | 64.11 |
| Average | 67.32 | 64.81 | 67.87 | 56.47 |
| Gain | – | -2.51 | 0.55 | -10.85 |

**Table 5:** Ablation study under the SL setting.

ment brought by the autoregressive decoder is marginal compared to the SGT module and the new features. Therefore, we conclude CPD module dramatically improves the evaluation speed while maintaining good recovery. (3) If we remove the introduced angular features, AEDesign is not as accurate as GCA and GVP, but the improvement in efficiency is still significant.

## 4.5 VISUAL EXAMPLES

We show the potential of AEDesign in real applications, i.e., designing all-alpha, all-beta, and mixed native proteins. We ensemble multiple models by selecting the sequence with the lowest perplexity as the final solution. We use AlphaFold2 to predict the structures of designed sequences and compare them with the reference ones. Visual examples are provided in Figure. 6.

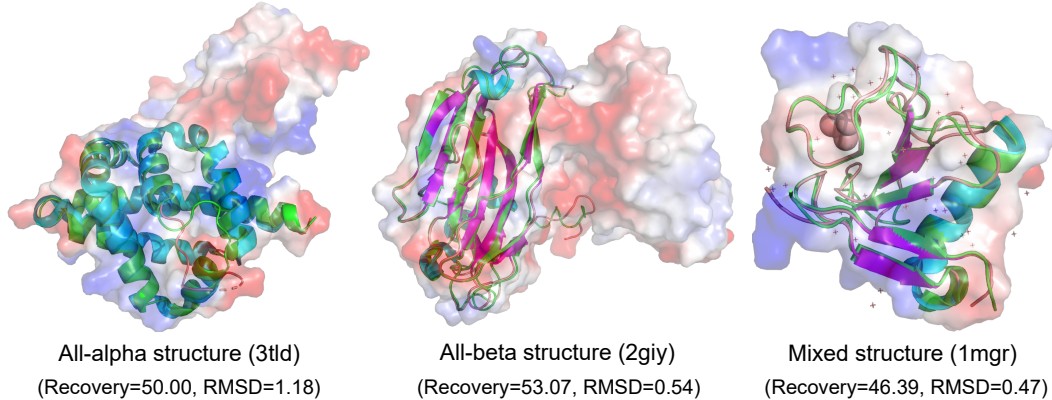

All-alpha structure (3tld)          All-beta structure (2giy)          Mixed structure (1mgr)

(Recovery=50.00, RMSD=1.18)     (Recovery=53.07, RMSD=0.54)     (Recovery=46.39, RMSD=0.47)

**Figure 6:** Visual examples. For native structures, we color Helix, Sheet, and Loop with cyan, magenta, and orange, respectively. Green structures are protein chains designed by our algorithm. We provide the recovery score and structural RMSD relative to the ground truth proteins.

## 5 CONCLUSION

This paper establishes a new benchmark and proposes a new method (AEDesign) for AI-guided protein design. By introducing new protein features, simplifying the graph transformer, and proposing a confidence-aware protein decoder, AEDesign achieves state-of-the-art accuracy and efficiency. We hope this work will help standardize comparisons and provide inspiration for subsequent research.

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

# A   DATA STATISTICS

**Protein length**   In Table.6, we count the protein in each species dataset by length. AlphaFold DB contains some extra-long proteins, so it is necessary to improve the algorithm efficiency.

| ID | Name | Structures | $\leq 30$ | $(30, 500)$ | $[500, 1000]$ | $> 1000$ |
|----|------|-----------|-----------|-------------|---------------|----------|
| 1 | DANRE | 24,664 | 27 | 15,460 | 6,573 | 2,604 |
| 2 | CANAL | 5,974 | 0 | 3,656 | 1,874 | 444 |
| 3 | MOUSE | 21,615 | 55 | 13,767 | 5,668 | 2,125 |
| 4 | ECOLI | 4,363 | 63 | 3,719 | 526 | 55 |
| 5 | DROME | 13,458 | 28 | 8,509 | 3,563 | 1,358 |
| 6 | METJA | 1,773 | 0 | 1,605 | 144 | 24 |
| 7 | PLAF7 | 5,187 | 1 | 2,832 | 1,313 | 1,041 |
| 8 | MYCTU | 3,988 | 4 | 3,365 | 536 | 83 |
| 9 | CAEEL | 19,694 | 52 | 15,073 | 3,592 | 977 |
| 10 | DICDI | 12,622 | 3 | 7,663 | 3,367 | 1,589 |
| 11 | TRYCC | 19,036 | 6 | 12,622 | 5,007 | 1,401 |
| 12 | YEAST | 6,040 | 24 | 3,789 | 1,715 | 512 |
| 13 | SCHPO | 5,128 | 5 | 3,385 | 1,386 | 352 |
| 14 | RAT | 21,272 | 12 | 13,884 | 5,370 | 2,006 |
| 15 | HUMAN | 23,391 | 49 | 12,399 | 5,653 | 5,290 |
| 16 | ARATH | 27,434 | 42 | 19,885 | 6,389 | 1,118 |
| 17 | MAIZE | 39,299 | 83 | 29,145 | 8,360 | 1,711 |
| 18 | LEIIN | 7,924 | 0 | 4,276 | 2,505 | 1,143 |
| 19 | STAA8 | 2,888 | 22 | 2,567 | 267 | 32 |
| 20 | SOYBN | 55,799 | 17 | 41,048 | 12,353 | 2,381 |
| 21 | ORYSJ | 43,649 | 78 | 35,987 | 6,738 | 846 |
| | Sum | 365,198 | 571 | 254,636 | 82,899 | 27,092 |

**Table 6:** AlphaFold DB: we show the total number of proteins $N_{all}$, and the number of proteins whose length within $(0, 30]$, $(30, 500]$, $(500, 1000]$ and $(1000, +\infty]$. The statistics of all species-specific subsets are also presented.

# B PREPROCESSING

**Filter low quality data by pLDDT**   Similar to ESM-IF (Hsu et al., 2022), we address data quality and partitioning issues through data pre-processing for each species-specific subset. As suggested by (Baek & Kepp, 2022; Callaway, 2020), the MAE between the predicted and experimentally generated structures does depend on pLDDT. Thus, we filter low-quality structures whose confidence score (pLDDT) is less than 70.

**Filter test data by sequence identity**   To prevent potential information leakage, for each species-specific subset, we cluster protein sequences if their sequence similarities (Qi & Zhang, 2020; Steinegger & Söding, 2017) are higher than 30% (Qi & Zhang, 2020) and split the dataset by these clusters. As a result, the proteins belonging to the same cluster must be in one of the training, validation, and test sets. By default, we keep the ratio of training set to test set around 9:1 and select 100 proteins from a randomly selected cluster for validation. If the randomly selected cluster has less than 100 proteins, then all of its proteins are used as the validation set. In Table.7, we show the number of training data (#Train), validation data (#Valid), and test data (#Test) under the partition based on 30% sequence identity.

**Filter test data by TS-score**   In addition to the sequence identity clustering, we further filter the test sets by structural similarity using Foldseek (van Kempen et al., 2022) to exclude any structures with TM-score larger than 0.5 from those in the training set. Thus, the training and test sets are strictly different at both the sequence and structure levels. In Table.7, we show the number of test data (#TMTest) after filtering by 30% sequence identity and 0.5 TM-score.

| ID | Name | SL setting | | | | SA setting | | | |
|---|---|---|---|---|---|---|---|---|---|
| | | #Train | #Valid | #Test | #TMTest | #Train | #Valid | #Test | #TMTest |
| 1 | DANRE | 13914 | 100 | 1446 | 493 | 22197 | 100 | 2367 | 808 |
| 2 | CANAL | 3290 | 99 | 267 | 168 | 5377 | 99 | 498 | 240 |
| 3 | MOUSE | 12417 | 73 | 1277 | 412 | 19524 | 28 | 2063 | 862 |
| 4 | ECOLI | 3347 | 100 | 272 | 114 | 3926 | 100 | 337 | 129 |
| 5 | DROME | 7672 | 86 | 751 | 369 | 12112 | 100 | 1246 | 610 |
| 6 | METJA | 1444 | 80 | 81 | 42 | 1595 | 88 | 90 | 47 |
| 7 | PLAF7 | 2561 | 87 | 184 | 105 | 4668 | 100 | 419 | 267 |
| 8 | MYCTU | 3028 | 100 | 237 | 98 | 3589 | 100 | 299 | 78 |
| 9 | CAEEL | 13565 | 95 | 1413 | 522 | 17724 | 96 | 1874 | 743 |
| 10 | DICDI | 6896 | 100 | 667 | 346 | 11359 | 100 | 1163 | 586 |
| 11 | TRYCC | 11360 | 98 | 1164 | 739 | 17132 | 86 | 1818 | 964 |
| 12 | YEAST | 3410 | 100 | 279 | 169 | 5436 | 100 | 504 | 263 |
| 13 | SCHPO | 3046 | 100 | 239 | 118 | 4616 | 99 | 413 | 200 |
| 14 | RAT | 12495 | 45 | 1344 | 56 | 19145 | 99 | 2028 | 858 |
| 15 | HUMAN | 11160 | 99 | 1140 | 452 | 21051 | 100 | 2240 | 955 |
| 16 | ARATH | 17897 | 99 | 1889 | 750 | 24690 | 94 | 2650 | 910 |
| 17 | MAIZE | 26232 | 98 | 2815 | 1220 | 35369 | 100 | 3830 | 1420 |
| 18 | LEIIN | 3848 | 100 | 328 | 201 | 7131 | 100 | 693 | 411 |
| 19 | STAA8 | 2310 | 100 | 157 | 76 | 2599 | 100 | 189 | 75 |
| 20 | SOYBN | 36944 | 99 | 4005 | 1732 | 50226 | 92 | 5481 | 2152 |
| 21 | ORYSJ | 32388 | 97 | 3502 | 2047 | 39289 | 95 | 4265 | 2273 |
| | Sum | 229224 | 1955 | 23457 | 10229 | 328755 | 1976 | 34467 | 14851 |

**Table 7:** Dataset splits. We show the number of training data (#Train), validation data (#Valid), and test data (#Test) under the partition based on 30% sequence identity. We also exhibit the number of test data (#TMTest) after filtering by 30% sequence identity and 0.5 TM-score.

# C    RESULTS AFTER TM-SCORE FILTERING

**TM-score filtered Results**    We evaluated all models on test sets filtered by both 30% sequence identity and 0.5 TM-score, and provide the TM-score filtered benchmarks on Table.8 and Table.9. Compared to the results without using the TM-score filter, the relative performance gain of our model is slightly reduced. The average model performance on this TM score-based test set remains nearly the same as on the sequence identity-based test set. We note that ESM-IF also finds that "the model performance overall remains the same on the TM score-based test set as on the CATH topology split test set." These facts show that more filters do not make the model predictions as difficult as we expected. This means that the model may not rely on the so-called homologous information leakage for prediction, but actually learns the patterns of the data.

**Table 8:** TM-score filtered SL benchmark. The length of the protein must be between 30 and 500. The sequence identity and TM-score between training and testing proteins are less than 30% and 0.5, respectively.

| | GTrans | SGNN | GVP | GCA | Our(SL) | Gain |
|---|---|---|---|---|---|---|
| DANRE | 42.86 | 53.57 | 60.17 | 61.96 | **69.18** | 7.22 |
| CANAL | 47.69 | 51.56 | 52.52 | 59.20 | **62.22** | 3.02 |
| MOUSE | 48.55 | 53.74 | 58.60 | 62.61 | **68.65** | 6.04 |
| ECOLI | 38.55 | 52.46 | 54.93 | 59.32 | **60.49** | 1.17 |
| DROME | 43.20 | 53.98 | 60.00 | 62.07 | **68.91** | 6.84 |
| METJA | 42.86 | 52.77 | 51.23 | 57.78 | **62.47** | 4.69 |
| PLAF7 | 48.00 | 50.49 | 51.59 | 61.29 | **64.10** | 2.81 |
| MYCTU | 53.88 | 56.42 | 58.08 | 66.45 | **69.20** | 2.75 |
| CAEEL | 48.42 | 56.05 | 63.65 | 66.67 | **74.24** | 7.57 |
| DICDI | 41.18 | 53.81 | 59.68 | 64.07 | **69.85** | 5.78 |
| TRYCC | 54.13 | 59.04 | 64.24 | 64.68 | **74.55** | 9.87 |
| YEAST | 34.86 | 49.28 | 51.95 | 58.82 | **62.07** | 3.25 |
| SCHPO | 46.35 | 49.32 | 52.17 | 59.77 | **62.43** | 2.66 |
| RAT | 44.14 | 54.43 | 59.52 | 63.05 | **70.96** | 7.91 |
| HUMAN | 50.29 | 54.48 | 61.06 | 63.83 | **70.69** | 6.86 |
| ARATH | 44.83 | 55.68 | 62.86 | 65.05 | **72.26** | 7.21 |
| MAIZE | 46.62 | 55.56 | 64.13 | 66.33 | **75.00** | 8.67 |
| LEIIN | 50.50 | 52.96 | 56.25 | 61.36 | **66.52** | 5.16 |
| STAA8 | 45.81 | 47.54 | 48.41 | 58.65 | **59.90** | 1.25 |
| SOYBN | 46.93 | 55.74 | 63.16 | 64.43 | **73.68** | 9.25 |
| ORYSJ | 45.14 | 54.12 | 62.50 | 64.10 | **73.77** | 9.67 |
| Average | 45.94 | 53.48 | 57.94 | 62.45 | **68.15** | 5.70 |

**Table 9:** TM-score filtered SA benchmark. There is no constraint on the protein length. The sequence identity and TM-score between training and testing proteins are less than 30% and 0.5, respectively.

| | GTrans | SGNN | GVP | GCA | Our(SA) | Gain |
|---|---|---|---|---|---|---|
| DANRE | 47.83 | 56.27 | **65.00** | 64.91 | 63.54 | -1.46 |
| CANAL | 51.59 | 54.84 | 62.02 | 61.67 | **69.26** | 7.24 |
| MOUSE | 47.06 | 56.99 | 63.64 | 64.77 | **74.06** | 9.29 |
| ECOLI | 50.41 | 54.02 | 59.43 | 61.23 | **66.11** | 4.88 |
| DROME | 50.97 | 56.46 | 63.55 | 62.93 | **72.88** | 9.33 |
| METJA | 51.24 | 52.50 | 55.60 | **59.80** | 59.79 | -0.01 |
| PLAF7 | 48.30 | 54.39 | 58.32 | 60.35 | **67.04** | 6.69 |
| MYCTU | 46.66 | 56.43 | 59.84 | 63.50 | **69.09** | 5.59 |
| CAEEL | 46.95 | 53.73 | 64.40 | 66.34 | **74.59** | 8.25 |
| DICDI | 56.85 | 58.10 | 64.86 | 64.82 | **73.51** | 8.65 |
| TRYCC | 50.26 | 61.41 | 66.36 | 69.08 | **77.73** | 8.65 |
| YEAST | 49.62 | 53.90 | 59.92 | 59.01 | **67.69** | 7.77 |
| SCHPO | 47.31 | 53.43 | 59.38 | 60.26 | **66.83** | 6.57 |
| RAT | 47.14 | 56.52 | 64.16 | 64.00 | **73.21** | 9.05 |
| HUMAN | 55.37 | 56.00 | 64.36 | 64.41 | **66.83** | 2.42 |
| ARATH | 47.03 | 56.38 | 65.04 | 64.70 | **73.82** | 8.78 |
| MAIZE | 56.46 | 58.05 | 66.04 | 66.67 | **76.00** | 9.33 |
| LEIIN | 55.64 | 58.00 | 64.55 | **65.03** | 64.62 | -0.41 |
| STAA8 | 48.68 | 51.35 | 53.85 | 58.95 | **63.16** | 4.21 |
| SOYBN | 48.98 | 57.52 | 64.60 | 69.85 | **76.78** | 6.93 |
| ORYSJ | 46.38 | 57.14 | 64.81 | 65.12 | **75.36** | 10.14 |
| Average | 50.03 | 55.88 | 62.37 | 63.69 | **70.09** | 6.40 |

# D  BENCHMARK ON CATH4.2

**How we re-run baselines**   We tune hyperparameters of all models on CATH4.2. For baseline models, we prefer to choose the hyperparameters recommended in their original paper to ensure that the results produced by our code are consistent with those reported, as shown in Table.10. For AEDesign, we also tune the hidden dimensions (128) and the number of layers (10 layers of GNN + 3 layers of CNN) in CATH4.2 to investigate whether it could achieve competitive results. When adapting to the new dataset, such as AlphaFold DB, we we fix hyperparameters of all models, including AEDesign, to be the same as those on CATH4.2. For each model, we use the same batch size as in the reproduction phase and adjust the learning rate in [ 0.001, 0.0001] to ensure that the model is converged stably.

**Results on CATH4.2**   We provide results where models are trained on CATH4.2 in Table.10 for investigating the potential of AEDesign model when designing native proteins. We use the same data splitting as GraphTrans (Ingraham et al., 2019) and GVP (Jing et al., 2020), where proteins are partitioned by the CATH topology classification, resulting in 18024 proteins for training, 608 proteins for validation, and 1120 proteins for testing. We observe that the non-autoregressive AEDesign outperforms its competitors in recovery while the autoregressive GVP and GCA achieve lower perplexity. Since recovery is the primary measure, we conclude that AEDesign is competitive in designing native proteins. More importantly, the performance order of models is consistent with the results on AlphaFold DB: AEDesign > GCA ≈ GVP > StructGNN > GraphTrans.

**Table 10:** Results comparison on the CATH dataset. All baselines are reproduced under the same code framework, where perplexity (lower is better) and recovery (higher is better) are reported. The **best** and next best results are labeled with bold and underline.

|  | Model | Perplexity ↓ | | | Recovery % ↑ | | |
|---|---|---|---|---|---|---|---|
|  |  | Short | Single-chain | All | Short | Single-chain | All |
| Reported | StructGNN | 8.33 | 8.86 | 6.55 | – | – | – |
|  | GraphTrans | 8.54 | 9.03 | 6.85 | – | 27.6 | – |
|  | GVP | 7.10 | 7.44 | 5.29 | 32.1 | 32.0 | 40.2 |
|  | GCA | 7.68 | 8.09 | 6.44 | 33.25 | 33.04 | 36.11 |
| Reproduced | StructGNN | 8.29 | 8.74 | 6.40 | 29.44 | 28.26 | 35.91 |
|  | GraphTrans | 8.39 | 8.83 | 6.63 | 28.14 | 28.46 | 35.82 |
|  | GVP | 7.23 | 7.84 | **5.36** | 30.60 | 28.95 | 39.47 |
|  | GCA | **7.09** | **7.49** | 6.05 | 32.62 | 31.10 | 37.64 |
|  | AEDesign | 7.32 | 7.63 | 6.30 | **34.16** | **32.66** | **41.31** |

# E    DOMAIN GENERALIZATION

This work does not study the domain generalization problem, which could be another research direction. However, it will be good for readers to know the difference between proteins created by AlphaFold2 and native ones.

**Discussion about Domain shifts**    We study the domain shifts between proteins created by AlphaFold2 and native proteins. Taking all the testing sets of AF2DB and CATH4.2 as examples, we statistics the distribution of angle features to study whether there are significant differences, as shown in Table.11. We also compare angle distributions of AF2DB and CATH4.2 in Figure.7. We observe that the angle distributions are quite similar but not the same between AF2DB and CATH4.2. The similar distribution means that the knowledge learned from AF2DB could be transferred into CATH4.2, while the difference may lead to performance degradation when transferring to different domains.

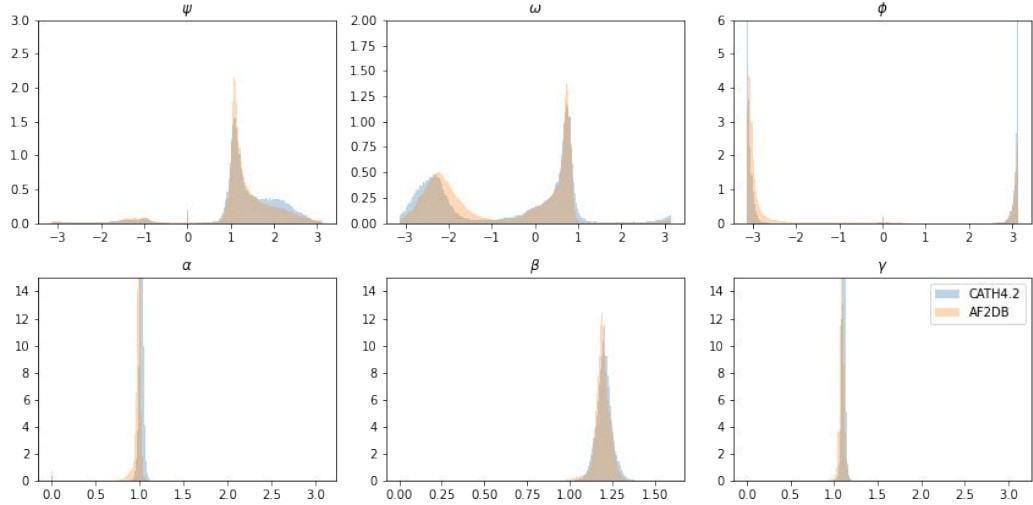

**Figure 7:** Comparing the angle distributions between CATH4.2 and AF2DB, structural noise = 0.00.

| Noise std | | $\psi$ | $\omega$ | $\phi$ | $\alpha$ | $\beta$ | $\gamma$ |
|---|---|---|---|---|---|---|---|
| | CATH4.2 | 1.34(0.89) | -0.66(1.57) | -0.40(3.04) | 1.01(0.09) | 1.20(0.05) | 1.11(0.08) |
| 0 | AF2DB | 1.20(0.96) | -0.78(1.44) | -1.53(2.55) | 0.98(0.07) | 1.19(0.05) | 1.09(0.07) |
| | KL | 1.38 | 3.10 | 17.12 | 175.64 | 11.76 | 52.46 |
| | CATH4.2 | 1.34(0.89) | -0.66(1.57) | -0.33(3.05) | 1.01(0.10) | 1.20(0.06) | 1.11(0.09) |
| 0.02 | AF2DB | 1.19(0.96) | -0.78(1.44) | -1.44(2.60) | 0.98(0.08) | 1.19(0.06) | 1.09(0.08) |
| | KL | 2.30 | 2.97 | 12.71 | 73.24 | 5.33 | 25.54 |
| | CATH4.2 | 1.34(0.90) | -0.65(1.57) | -0.22(3.03) | 1.02(0.12) | 1.21(0.09) | 1.11(0.11) |
| 0.05 | AF2DB | 1.19(0.97) | -0.78(1.44) | -1.16(2.72) | 0.98(0.11) | 1.19(0.09) | 1.09(0.10) |
| | KL | 1.07 | 2.61 | 9.25 | 18.21 | 1.87 | 3.58 |
| | CATH4.2 | 1.33(0.92) | -0.63(1.59) | -0.14(2.99) | 1.03(0.18) | 1.21(0.15) | 1.12(0.18) |
| 0.1 | AF2DB | 1.19(0.99) | -0.77(1.46) | -0.83(2.80) | 0.99(0.18) | 1.20(0.15) | 1.10(0.17) |
| | KL | 1.26 | 2.07 | 5.21 | 4.11 | 0.81 | 0.49 |

**Table 11:** Statistics of angle features. We count angle distributions for the testing sets of AF2DB (SL setting) and CATH4.2. The mean and standard deviation are provided, where the standard deviation is marked in brackets. We also provide KL divergence between the angle distributions of AF2DB and CATH4.2. Gaussian noise could be added to the input structures, where the noise std is listed in the left.

**Address the domain shift issue**    How to eliminate domain differences? Inspired by Li at.el(Li et al., 2021b; Dauparas et al., 2022; Hsu et al., 2022), we find that perturbing input structures with Gaussian noise during training leads to improved domain-generalization performance. As shown in Table.11, the KL divergence between features of AF2DB and CATH4.2 decreases when the the standard deviation of Gaussian noise increases. When training AEDesign on SOYBN (a subset of

AF2DB) and evaluating it on the testing set of CATH4.2, the generalization can be enhanced by adding noise, as shown in Table.12. However, this does not mean that higher noise levels are better, as too much noise may obscure all useful information.

| Noise std | 0.00 | 0.001 | 0.02 | 0.05 | 0.08 | 0.10 | 0.20 | 0.30 |
|---|---|---|---|---|---|---|---|---|
| CATH4.2 | 20.74 | 29.21 | 32.62 | 34.51 | 34.24 | 33.63 | 31.90 | 29.26 |

**Table 12:** Results under the cross-domain setting. We train AEDesign on the training set of SOYBN, and evaluate it on the testing set of CATH4.2. We reveal how the noise level helps the domain generalization.

We have discussed the domain shift issue between AF2DB and CATH4.2, and verified that the model trained on AF2DB by simply adding Gaussian noise is effective for designing native proteins. This discovery is consistent with the results of ESM-IF and ProteinMPNN. However, we provide new insight into why adding noise is effective. We believe that better domain generalization methods could further improve the model's performance, but we do not intend to study it in depth in this work. Moreover, ESM-IF has verified that: training AF2DB with CATH4.2 should further improve the model performance from 38.3% to 51.6% on the CATH dataset, and this paper does not repeat their innovation. From our perspective, ESM-IF's data augmentation is actually another domain generalization approach.

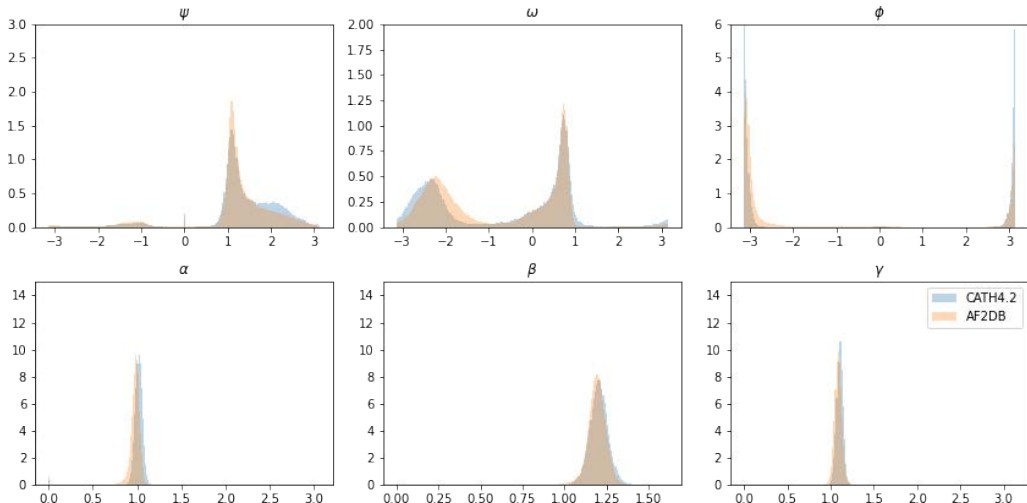

**Figure 8:** Comparing the angle distributions between CATH4.2 and AF2DB, structural noise = 0.02.

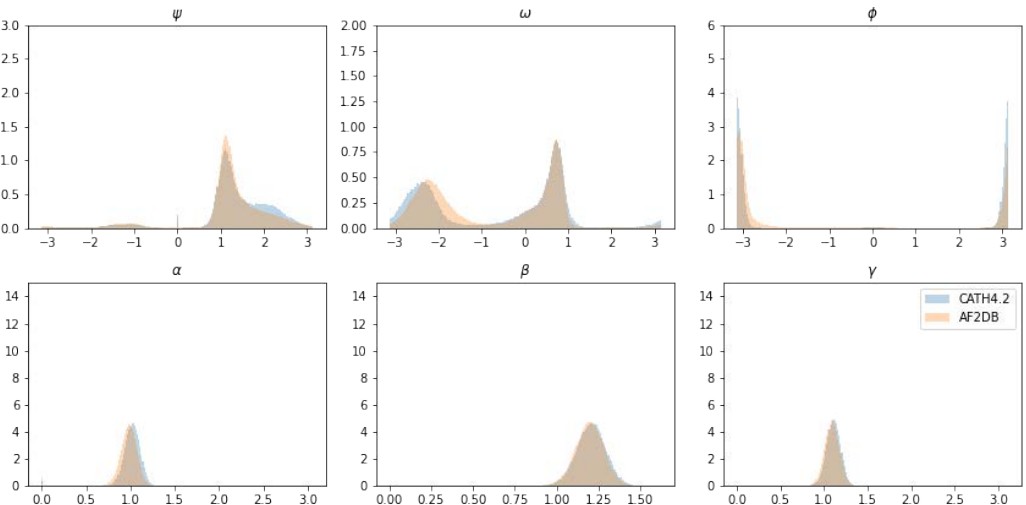

**Figure 9:** Comparing the angle distributions between CATH4.2 and AF2DB, structural noise = 0.05.

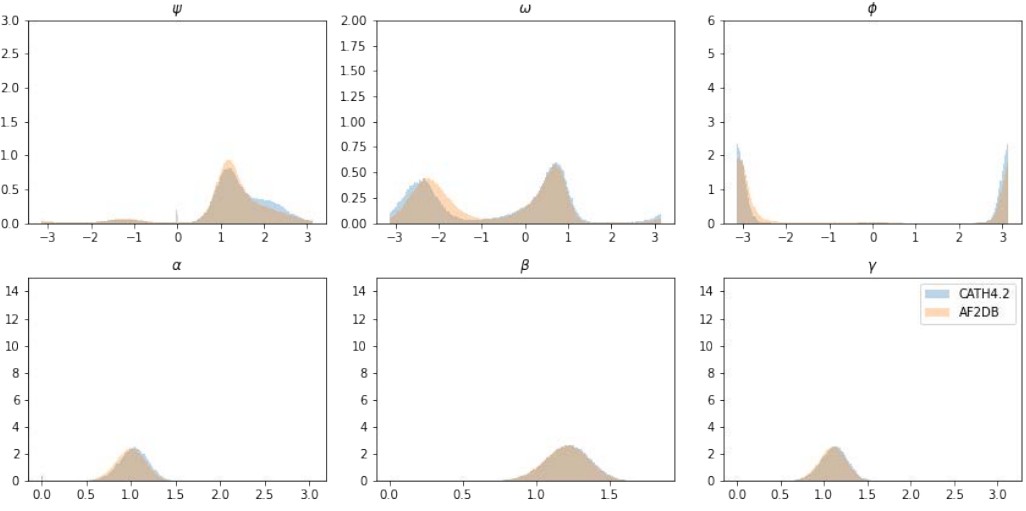

**Figure 10:** Comparing the angle distributions between CATH4.2 and AF2DB, structural noise = 0.10.

