# OpenReview forum: "AlphaDesign: A graph protein design method and benchmark on AlphaFold DB"
_ICLR.cc/2023/Conference — Submitted to ICLR 2023_

### Official Review · Reviewer_QZ9p · 2022-10-21

**Confidence:** 4
**Correctness:** 4
**Technical Novelty And Significance:** 2
**Empirical Novelty And Significance:** 3
**Recommendation:** 6

**Clarity, Quality, Novelty And Reproducibility:**

The paper is well written and approachable.

One of the missions of the paper is to provide a new dataset and train-test split for benchmark that will result in better reproducible research in the future.

The overall modeling approach leverages standard techniques from modern deep learning, but the empirical results are strong.

**Strength And Weaknesses:**

=Strengths=
Interesting/relevant application
Strong empirical results
Establishes a benchmarking dataset to be used in future papers.
Useful ablations to identify the impact of various aspects of the proposed method.

=Weakness=
I have a few key hesitations around the comparison to recent work, how the dataset was constructed, etc. See my extended comments below.


**Summary Of The Paper:**

The paper proposes a new neural network architecture for the inverse folding problem (protein structure -> sequence) and develops a new benchmarking setup that the authors recommend for future work in the field. The new benchmark improves on prior practices by formalizing a train-validation-test split and by providing more diversity of proteins. The empirical performance of the proposed model is strong relative to prior work.

**Summary Of The Review:**

I appreciate that the paper works to introduce a new benchmarking setup. This kind of work can be under-appreciated and is important to drive progress in the field. I also appreciate that the performance of the proposed method is strong.

I have some key questions below:

The benchmark is based on a fairly old version of AlphaFoldDB. The current database contains ~200M structures. Using this would allow you to perhaps have a more diverse test set and to guarantee a bigger distance between the train and test sets. Why did you use an old version of the database?

I don't understand why the results are stratified by organism. How does this improve the analysis? Homologous proteins will be fairly similar across different organisms. If you were to stratify, why not stratify by type of molecular function?

Also, when you constructed the clustering-based train-test split, did you cluster proteins from multiple organisms, or did you do clustering independently for each organism?

I'm concerned that the paper does not benchmark against recently published papers on this topic. The paper cites the ESM-IF paper, for example, but explains that no benchmarking was performed because the work was in parallel and code has not been released. However, the paper was accepted at a prior ML conference (ICML 2022) and the code was released [1] about 4 months ago. Can you elaborate on why you do not consider the method? Is the challenge that ESM-IF would need to be retrained because you use a different train-test split?
[https://github.com/facebookresearch/esm/tree/main/examples/inverse_folding]

One challenge of introducing a new benchmark dataset is that you have to re-run baseline methods. What protocols did you use, for example, when adapting GraphTrans, etc. to the new data? Did you re-tune any hyper-parameters?


==Update after authors' response==
I have raised my score to weak accept. My concern around technical correctness (the per-organism train-test split) has been adequately clarified. I wish the paper used the full AlphaFoldDB, but it would not be fair to penalize this paper when the AlphaFoldDB was updated not long before the ICLR deadline.

---

### Official Review · Reviewer_MXiS · 2022-10-22

**Confidence:** 4
**Clarity, Quality, Novelty And Reproducibility:** The work is clear, but several concur…
**Correctness:** 2
**Technical Novelty And Significance:** 2
**Empirical Novelty And Significance:** 2
**Recommendation:** 1

**Strength And Weaknesses:**

Strengths:
- AlphaDesign's model is interesting, and does seem much faster than decoding autoregressively, especially on long sequences.
- The paper comprehensively tests a variety of different models, showing that AlphaDesign produces good designs across species.

Weaknesses:
- The new dataset is unsound, it presumes that AlphaFoldDB is producing ground truth structures that can be used as a validation set. I recommend authors make an argument that switching to alphafold structures has no impact on design of natural proteins, otherwise it would be akin to testing on a synthetic dataset that might not transfer to real protein data.
- It seems unreasonable not to include _any_ natural proteins in this work.
- There's no validation set used, which means models may be overfitting on the test set, especially if the authors did more tuning on their model than the baselines.
- The dataset is not held out in a structural manner. In Hsu et al, 2022; Ingraham et al., 2019; Jing et al., 2021b; Strokach et al., 2020, they all use CATH based topology splits to ensure that the result holds up across new topologies. One way to do this might be to use TMscore to do clustering of training / test sets.

**Summary Of The Paper:**

- The paper establishes a new benchmark on fixed backbone protein design using AlphaFold DB
- Authors introduce a new model based on a simplified graph transformer encoder and a constraint aware decoder
- This model achieves good results in comparison to a large number of baseline models using GVP or graph transformer.

**Summary Of The Review:**

The paper does not use current best practices for data splits, and also trains/tests on a synthetic database without showing that there's no domain shift issues. Additionally, many ideas have been implemented in concurrent work. Therefore, I recommend a reject rating for this work.

---

### Official Review · Reviewer_dT5A · 2022-10-24

**Confidence:** 5
**Clarity, Quality, Novelty And Reproducibility:** See Strength And Weaknesses.
**Correctness:** 3
**Technical Novelty And Significance:** 3
**Empirical Novelty And Significance:** 3
**Recommendation:** 6

**Strength And Weaknesses:**

Pros:
1. The motivation is clear.
2. The paper is well-written and organized.
Cons:
1. The main contributions are not clear.
2. Some related works are missing, e.g., Multi-Human Parsing With a Graph-based Generative Adversarial Model.

**Summary Of The Paper:**

This paper establishes a new benchmark based on AlphaFold DB, one of the world’s largest protein structure databases. Moreover, the authors propose a new baseline method called AlphaDesign, which achieves 5% higher recovery than previous methods and about 70 times inference speed-up in designing long protein sequences. The authors also reveal AlphaDesign’s potential for practical protein design tasks, where the designed proteins achieve good structural compatibility with native structures.

**Summary Of The Review:**

See Strength And Weaknesses.

---

### Official Review · Reviewer_QTBS · 2022-10-24

**Confidence:** 4
**Correctness:** 3
**Technical Novelty And Significance:** 2
**Empirical Novelty And Significance:** 4
**Recommendation:** 6

**Clarity, Quality, Novelty And Reproducibility:**

## Clarity
Generally, the paper is well-structured, with clear illustrations. However, the method section shows signs of being hastily written, and would benefit from some editing (see detailed comments below).

## Quality
The work seems well thought through and executed. The manuscript could use a bit more polishing and perhaps a supporting information  appendix with some extra details about the method.

## Reproducibility
The manuscript stresses the importance of open source and promises to provide source code when released. It also includes a benchmark with the goal of improving reproducibility in the field. I therefore trust that the results of this paper should be readily reproducible.

## Detailed comments
Title. The title suggests that this project originates from DeepMind, following their naming style. If this paper is by another group, I would suggest that the authors change the name, to avoid confusion between AlphaFold and AlphaDesign. Especially since the proposed procedure does not seem to be methodologically related to AlphaFold in any way - other than using the structures provided by Alphafold for training.

Page 1. "may have overlooked some important protein features" and "few of them exceeds 50% accuracy". These are odd statement to make in an introduction about existing work if you do not identify which features you think are missing, and you haven't yet introduced which task you are considering (50% accuracy of what?). Both things become clearer later in the paper, but I would recommend removing these statements here.

Page 2. "length-free" (also used several other places)
Length-free is an odd term, suggesting that the proteins have no length. Consider rephrasing to something like "arbitrary lengths".

Page 2. Related work.
The authors seem to have missed some of the earlier CNN work on inverse folding:
https://pubmed.ncbi.nlm.nih.gov/28615003/

https://proceedings.neurips.cc/paper/2017/hash/1113d7a76ffceca1bb350bfe145467c6-Abstract.html
https://proceedings.neurips.cc/paper/2018/hash/488e4104520c6aab692863cc1dba45af-Abstract.html

Page 2. "Sovlent-accessible" -> "Solvent accessible"

Page 3. "none of the above 3D CNN models is open-source"
Several of the papers linked to above provide source code.

Figure 1:
The figure says "Constraint-aware Protein Decoder" while the caption says "confidence-aware" protein decoder. Is this a mistake?

Figure 4: "remains exploring"
Rephrase. E.g. “is an open research question.” or “is subject to further research”.

Figure 5. Section 3.3
This section is not very clear. Some examples:
1. The "Confidence Predictor" in Figure 4 takes a feature vector as input, while the "Conf" function in eq 3 takes a structure X as input. Are these different functions? I guess so, since Conf(.) is thereafter explained as "the model containing graph encoder and CNN decoder" - but this should be stated more clearly.
2. It is stated that f(.) computes confidence scores, but not what input it takes. I assume z?
3. \hat a is used without being defined.
4. There is a very odd footnote, which should be integrated into the main text.
5. "By extending \hat a as a" Eq (5)". It is not clear what "extending" means - and Equation 5 does not seem to exist.

As a result, it is unclear how this part of the model works. Since this is the main methodological contribution of the paper, the authors should rewrite and expand this section (if space is an issue, perhaps in supporting information).

Page 6
"systemical" -> "systematic"

Page 6. "This dataset provide well-organized proteomic data".
The term "proteomics" is usually used to describe large-scale studies of proteomes. It is therefore a bit confusing when you use it here to describe the AlphaFold protein structure database. I would recommend choosing a different term (you use it several places further down in the manuscript as well).

Page 6. "each proteomic data" x2
Do you mean species-specific subsets here? Again, "proteomic" is confusing (see above).

Page 7. "Length-free"x2. Rephrase

Page 7. "each proteomic data". Rephrase

Table 2. Caption. Do you mean "next best" when you write "suboptimal"? Considering rephrasing.

Page 9. Visual examples. It is not entirely clear what the relevance is of this test. Are you testing AlphaFold or AlphaDesign with this test - or their internal consistency?

**Strength And Weaknesses:**

The paper makes good use of the recently released AlphaFold Database of protein structures, both by training a new model on these available structures, and by curating the database into a dataset which can be used as a standard for future work. Method-wise, the contribution is limited - the authors build on components from earlier work, adding three new angle features and a new decoder. However, the results are convincing and the method is therefore likely to have an impact.

Overall, the paper is clearly structured. However, parts of the paper seem hastily written and in particular parts of the Methods section are not clearly explained. See detailed comments below.


**Summary Of The Paper:**

The manuscript introduces a new method for protein design (inverse folding) and a benchmark consisting of a curated set of structures from the AlphaFold Database. The authors demonstrate a substantial improvement compared to a selection of earlier methods.

**Summary Of The Review:**

The paper presents a new method for inverse folding.  The main strength of the paper lies in the reported results, which look very competitive - and in the fact than the authors provide a new benchmark to the community. The manuscript does not contain a substantial methodological contribution, and the part that is new is currently not described very clearly in the paper. However, based on the results I expect the paper could have an impact on the community. If the authors improve the clarity of the paper by addressing the concerns described in the detailed comments above, I would therefore be willing to increase my score.

---UPDATE---
The authors have improved the clarity of the paper as requested, and I have therefore increased my score to 6: marginally above the acceptance threshold.

---

### Decision · Program_Chairs · 2023-01-20

**Decision:**

Reject

**Justification For Why Not Higher Score:**

The contributions are incremental and not novel enough. There are serval ICLR submission on this topic which are much more novel and which would greatly benefit the community.

**Justification For Why Not Lower Score:**

N/A

**Metareview: Summary, Strengths And Weaknesses:**

The paper focuses on the protein inverse folding problem and proposes a new approach which introduces new protein features and simplify GVP to improve accuracy and efficiency, and a new benchmark leveraging the AlphaFold Protein Structure Database.

The AC is very familiar with the particular problem and carefully examined both the manuscript and feedback. The AC and reviewers appreciate the clarifying elements and additional material provided by the authors during the discussion phase. As a result, some reviewers raised their scores to weak accept. However,  all agree that the contributions are of limited novelty and none can strongly feel in favor of acceptance. We therefore very much encourage the authors to resubmit their work to a conference which will appreciate the balance between new benchmark and methodological development.